# Phylogenetic Analysis of Bovine Respiratory Syncytial Virus (BRSV) Subgroups in Wallonia Region of Belgium in Relation to Current Vaccination Strategies

**DOI:** 10.3390/vaccines13030298

**Published:** 2025-03-11

**Authors:** Anneliese Demil, Mickaël Dourcy, Mutien-Marie Garigliany, Dominique Cassart, Fabien Grégoire, Julien Evrard, Daniel Desmecht, Calixte Bayrou, Hani Boshra

**Affiliations:** 1Department of Morphology and Pathology, Fundamental and Applied Research for Animals & Health (FARAH), Faculty of Veterinary Medicine, University of Liège, Bât B43, 4000 Liege, Belgium; a.demil@uliege.be (A.D.); mickael.dourcy@uliege.be (M.D.); mmgarigliany@uliege.be (M.-M.G.); dominique.cassart@uliege.be (D.C.); daniel.desmecht@uliege.be (D.D.); 2Regional Association for Animal Registration and Health (ARSIA) asbl, 5590 Ciney, Belgium; fabien.gregoire@arsia.be (F.G.); julien.f.evrard@gmail.com (J.E.); 3Bovine Health Service, Clinical Department of Production Animals (FARAH), Faculty of Veterinary Medicine, University of Liège, Bât B42, 4000 Liege, Belgium

**Keywords:** BRSV, livestock diseases, respiratory diseases, phylogenetics

## Abstract

**Background:** Bovine respiratory syncytial virus (BRSV) is a major pathogen of the bovine respiratory disease complex and causes regular severe winter outbreaks of respiratory disease in cattle. It is therefore responsible for important economic losses for the farming industry. In this study, the genetic diversity of the circulating BRSV strains in Belgium, which has not been assessed since the end of the 1990s, was investigated. **Methods**: We analyzed 51 BRSV-positive samples collected from 2015 to 2023. This study is the first report on the circulation of BRSV subgroup VIII in Belgium. Furthermore, co-circulation of subgroups II and III was recorded in the same period. Four commercially available vaccine strains marketed in Belgium were also included in the analysis and they clustered with subgroup II or III. **Results**: Our findings indicate that different strains of BRSV are circulating in Belgium, including those from subgroups II and VIII, with the subgroup VIII strains being particularly distant from the commercially available vaccine strains. **Conclusions**: These results highlight the importance of ensuring that the available vaccines efficiently protect against strains from circulating subgroups and assessing the potential circulation of attenuated vaccine strains.

## 1. Introduction

Bovine respiratory syncytial virus (BRSV or Bovine *Orthopneumovirus*) is a member of the *Orthopneumovirus* genus within the *Pneumoviridae* family. It is a major pathogen of the bovine respiratory disease complex in beef and dairy calves. The peak incidence of severe disease concerns mainly calves younger than 6 months of age [1,2] and it causes regular severe winter outbreaks of respiratory disease in cattle [3]. BRSV is closely related to the human respiratory syncytial virus (HRSV), the most important cause of lower respiratory tract disease in children [4]. BRSV and HRSV share common epidemiological, pathological and clinical characteristics [5]. Originally discovered in Europe [6], BRSV is now distributed worldwide [7,8,9]. Its impact on the cattle industry is associated with economic losses as a result of morbidity, mortality, costs of treatment and prevention, loss of production, and reduced carcass value [10]. Four modified-live virus (MLV) vaccine strains against BRSV are available in Belgium and were included in this study (Bovalto^®^ Respi Intranasal, Boehringer Ingelheim; Nasym^®^, Hipra; Bovilis^®^ Intranasal RSP, Intervet and Rispoval^®^ RS + Pi3 IntraNasal, Zoetis).

The genomic heterogeneity of the negative-stranded RNA BRSV genome and its low fidelity in terms of replication are important drivers of viral evolution and escape from vaccine protection [11]. Therefore, the analysis of BRSV strain diversity is essential for the development and update of efficacious vaccines. The gene encoding the attachment glycoprotein G has the highest reported mutation rate among the BRSV strains and is commonly used as a target for phylogenetic analyses [5]. Based on the sequence variability of the G open reading frame (ORF) within essential immunodominant regions, BRSV has been classified for a long time into six different genetic subgroups [5,12,13]. With better surveillance of BRSV circulation and given its high prevalence around the world, the number of subgroups has steadily increased; over the past two decades, the number of BRSV subgroups has ranged from six to eight subgroups [14,15], and more recently, to ten subgroups [16]. An extensive study of the diversity of the circulating BRSV strains in Europe is therefore needed to characterize the up-to-date circulation of distinct genetic subgroups on the continent.

In Belgium, the genetic diversity of the circulating BRSV strains has not been assessed since the end of the 1990s. The Wallonia region of southern Belgium plays an important role in the overall European livestock industry, with one of the highest livestock densities, being the birthplace of the Belgian Blue breed and occupying a central spatial position in Western Europe, sharing borders with France, Germany, Luxembourg and the Netherlands—all of which are major livestock producers on the continent [17]. An outbreak of any livestock disease in Wallonia would therefore have high potential to spread to these countries.

The aim of this study was to provide an update on the genetic diversity of the circulating BRSV strains in the Wallonia region of Belgium.

## 2. Materials and Methods

### 2.1. Field Samples

The objective of this study was to collect as many BRSV-positive samples as possible, either from the lungs of deceased animals or via bronchoalveolar lavage. The collection was conducted in collaboration with the necropsy room of the University of Liège and the laboratory of the Regional Association for Animal Identification and Health (ARSIA) between 2015 and 2023. Along with the age of the animals detected as positive for the virus, information on their vaccination status was collected.

Lung tissue samples or nasal swabs collected from suspected BRSV-infected animals were stored at −80 °C until thawed at room temperature for RNA extraction to confirm the BRSV positivity. The samples were analyzed using a specific RT-PCR assay to detect a 541 bp fragment of the BRSV G glycoprotein gene following previously described protocols [5,18].

### 2.2. BRSV Vaccines

To evaluate the phylogenetic relationship between the circulating viral strains and the vaccine strains, we included the vaccine strains in our analysis to determine their positioning relative to the circulating strains.

Four modified-live viral vaccine strains marketed in Belgium were obtained from vaccine bottles: MLV-A, MLV-B, MLV-C and MLV-D.

### 2.3. Extraction of BRSV RNA

For each sample, 50 mg of lung tissue or 200 μL of nasal swab preservation fluid was set in a 2 mL Eppendorf tube, along 500 μL of Nucleozol solution (Macherey-Nagel Inc., Düren, Germany). The tubes with lung tissue samples were homogenized with a 5 mm steel bead (Qiagen GmbH, Hilden, Germany) using the Qiagen Tissuelyser II device (3 min cycle at a frequency of 30 Hz). The homogenate of all the samples was centrifuged, with the supernatant being removed for subsequent RNA purification using the Nucleozol RNA extraction kit (Macherey-Nagel Inc., Düren, Germany), following the manufacturer’s instructions.

RNA from 3 samples with low-quality extraction (B22.0873, B19.223592 and B21.0944) was isolated using the TANBead^®^ Nucleic Acid Extraction kit (Taiwan Advanced Nanotech Inc., Taoyuan, Taiwan) according to the manufacturer’s instructions.

### 2.4. RT-PCR Targeting the BRSV G Gene

As demonstrated in multiple studies [5,19], the BRSV G glycoprotein exhibits the highest rate of genetic variation and is therefore the primary target of phylogenetic analyses. Primers corresponding to a 541 bp fragment of the BRSV glycoprotein (G) open reading frame were used, as previously described [5]. The primers used were as follows (5′ to 3′): VG1, CCACCCTAGCAATGATAACCTTGAC; VG4, GCTAGTTCTGTGGTGGATTGTTGTC. First-strand cDNA synthesis was generated using the Luna^®^ Universal Probe One-Step RT-qPCR Kit (New England Biolabs., Ipswich, MA, USA), where 10–15 μg of total extracted RNA was incubated with 0.4 μM of both VG1 and VG4 primers, using reaction buffers and enzymes according to the manufacturer’s instructions. The amplification conditions were 55 °C for 20 min at RT, denaturation at 95 °C for 5 min, followed by 50 cycles at 95 °C for 30 s, 55 °C for 30 s, and 72 °C for 60 s and a final elongation at 72 °C for 5 min. A second set containing one designed primer (Geneious 10.2.6 software, Biomatters Ltd., Auckland, New Zealand) corresponding to a 177 bp fragment was applied to 11 samples to extend the 5′ end. The primers used were as follows (5′ to 3′): FNew CCAAACACTCCCCYATGTGCCTTG associated with the same VG4. We increased the primer hybridization temperature to 62 °C to accommodate the minimum temperature requirement while keeping the rest of the amplification conditions unchanged. The PCR reactions were then run on a 1% agarose gel, stained with a fluorescent reagent (Midori Green Advance; Nippon Genetics), and detected under blue–green LED light. Bands corresponding to the theoretical molecular weight (i.e., 541 bp or 177 bp, respectively) were cut out of the gel for DNA purification (Figure 1).

### 2.5. DNA Purification and Sequencing of BRSV G Gene

The amplicons were gel-purified using the Nucleospin Gel and PCR Clean-Up kit (Macherey-Nagel Inc., Düren, Germany), following the manufacturer’s instructions. Sanger sequencing was performed by Eurofins Genomics, using the VG1/FNew and VG4 primers. The nucleotide sequences used in this study have been submitted to GenBank and assigned the accession numbers PP538078–PP538081 and PP538083–PP538133.

### 2.6. Phylogenetic Analysis

All the partial G ORF cDNAs were aligned by the ClustalW multiple alignment tool using the Geneious 10.2.6 software (Biomatters Ltd., Auckland, New Zealand). The experimental and vaccine sequences were compared with reference sequences of BRSV subgroups I–VIII selected from GenBank. All the experimental and reference sequences were equally trimmed to 418 bp. A reference sequence from ovine respiratory syncytial virus was used as an outgroup for the subsequent phylogenetic analyses.

The phylogenetic tree was generated with the MegaX v10.2.4 software package [20], using the maximum likelihood method with the Tamura–Nei distance with the gamma distribution, as determined by a model prediction analysis. The analysis was performed on untranslated cDNA sequences with 1000 bootstrap replications.

### 2.7. Amino Acid Sequence Analysis

The deduced amino acids from the major immunodominant zone (aa^159^–aa^186^) of both the experimental and reference sequences were aligned in Geneious 10.2.6 software (Biomatters Ltd.). The residues at positions 159–186 correspond to the central conserved region of the extracellular of the BRSV G glycoprotein, which is an important antigenic site [21].

### 2.8. Statistical Analysis

The proportion of vaccinated animals was compared between the main genetic BRSV subgroups using a Z-test for two proportions (Excel, Microsoft 365, v2408).

## 3. Results

### 3.1. Sample Collection

Lung tissue samples (*n* = 48) or nasal swabs (*n* = 3) from 51 BRSV-infected cattle were obtained from the “laboratory and diagnosis” department of the Regional Association for Animal Health and Identification (*n* = 17) and from the necropsy room of the Veterinary Faculty of Liège (*n* = 34) (Table 1). The mean age of our cohort was 5.5 months (standard deviation: 4.2 months).

The vaccination status was obtained for 42 animals. Twenty-one (50%) were vaccinated. For subgroup II, the proportion of vaccinated animals was 10/17 (58.8%). The proportion was lower for subgroup VIII, reaching 10/24 (41.6%). However, the Z-test applied to these two proportions showed no significant difference (*p*-value = 1.09 > 0.05).

Additional information, including the origin of the sample, sex, breed, and herd location, is provided in Appendix A.

### 3.2. Sequencing Results

The virus isolates belong to three different phylogenetic subgroups in Wallonia (II, III and VIII). The majority (94.1%) of samples were isolated from lung tissue after necropsy. The age and vaccine status of the animals and the accession numbers of the sequences are reported in Table 1.

### 3.3. Phylogenetic Classification

A phylogenetic tree was constructed using the maximum likelihood method to allow for comparison and clustering of the nucleotide sequences of the experimental, vaccine and reference isolates (Figure 2).

Almost all the field isolates were phylogenetically clustered with the reference strains from either subgroup II or subgroup VIII; out of the 51 field isolates, 20 clustered with the subgroup II reference sequences, while 30 isolates clustered with subgroup VIII. One field isolate (B19.207920) clustered with the reference sequences from subgroup III.

Among the four live-attenuated vaccine strains included in this study (Table 1), two vaccines clustered with the subgroup II reference sequences (MLV-A and MLV-B), while the other two (MLV-C and MLV-D) clustered with subgroup III. The MLV-D vaccine strain and the B19.207920 experimental isolate seemed to be phylogenetically very close, with only one nucleotide substitution (silent substitution).

### 3.4. Amino Acid Sequences

The translated sequences of the cloned BRSV cDNAs were analyzed for variability at specific amino acid positions within the major immunodominant zone located in the central conserved region of ectodomain (aa^159^–aa^186^) (Figure 3).

The four cysteine residues of the conserved central hydrophobic part of the ectodomain of the G glycoprotein of all the Belgian BRSV strains were conserved (i.e., positions 173, 176, 182 and 186) [11]. Some mutations were found within the central conserved region of some experimental and reference samples from different subgroups. Among the samples from subgroup II, the glutamate at position 177 was replaced by a lysine residue (E^177^ → K^177^) and the leucine at position 183 replaced by serine (L^183^ → S^183^). The mutations found within subgroup VIII were as follows: P^156^ → S^156^, H^162^ → Q^162^, N^163^ → T^163^ and S^165^ → L^165^. Concerning the vaccine strains (Table 1), several variations were observed within the immunodominant region. The MLV-C strain contained phenylalanine at position 165. As described for the field strains, reference strains and MLV-A vaccine strain from subgroup II, the glutamate at position 177 was replaced by a lysine (E^177^ → K^177^). On the contrary, the other subgroup-II-related MLV-B vaccine strain retained glutamate at this position (Figure 3). It should be noted that the MLV-C vaccine strain and the isolate strain B19.207920 belonging to subgroup III differed by only one amino acid (T^102^ → G^102^).

## 4. Discussion

The genetic diversity of the circulating BRSV strains in Belgium over a period of eight years (2015–2023) was analyzed based on the sequence coding for the extracellular domain of the glycoprotein G. Interestingly, the results of our phylogenetic analyses confirmed the current classification of BRSV strains into eight subgroups, as observed recently elsewhere in Europe [15]. Nevertheless, aside from recent research conducted in Croatia [15], Italy [14], Norway [13], and Sweden [19], there is a notable absence of up-to-date phylogenetic studies on BRSV across other European regions. Croatia is the only European country where subgroup VIII has also been detected: among the 24 samples analyzed, the strains were categorized into subgroups VII (14 out of 24) and VIII (9 out of 24). In Italy, an examination of 23 samples revealed the predominance of subgroup VII. In Sweden, 30 samples were analyzed, with subgroup II being the most prevalent. Similarly, in Norway, a study of 12 samples also indicated the predominance of subgroup II. The sequences of subgroups IX [22] and X [16], which have been described, respectively, in Brazil (10 samples) and Japan (22 samples), were not included in our analysis due to their short size (318 bp).

A majority of the strains (58.8%) clustered with subgroup VIII, a subgroup recently described on the European territory, more particularly in Croatia, but with a lower prevalence (37.5%; 9/24) [15]. The circulating BRSV strains in Belgium belong to the new subgroup VIII identified herein for the first time. This genetic evolution of BRSV could be heightened by herd immunity and in particular by the vaccine pressure. It has already been suggested that the BRSV strains from countries with intensive vaccination appear to evolve faster [5]. The BRSV vaccine programs are quite intensive in Belgium [5]. However, although the population of our study is limited, the proportion of vaccinated animals in the different genetic subgroups in this study did not support the hypothesis of a subgroup II mainly found in non-vaccinated animals and a subgroup VIII, which evades immunity more frequently, found in vaccinated animals.

Phylogenetic analysis of the vaccine strains available on the Belgian market indicates that they belong to either subgroup II or III. Considering that the majority of field samples were classified as subgroup VIII, this is of particular concern, as it is currently not known whether subgroup VIII is antigenically related to subgroup II or III. One implication of the hypothesis that subgroup VIII can evade vaccine-induced immunity is that the proportion of vaccinated calves might be higher in this subgroup compared to subgroup II. The latter could be predominantly found in non-vaccinated animals. However, our study found no significant difference in the proportion of vaccinated animals between the two groups. From that perspective, our data do not support the notion that subgroup VIII has the ability to evade immunity conferred by the currently available vaccines. Further studies are required to determine whether the immunity generated by using these vaccines can efficiently protect the host from an infection with the subgroup VIII strain of BRSV.

In addition, 39.2% of field strains belonged to subgroup II, which also contains older Belgian strains (≤1983) and two vaccine strains currently marketed. Contemporary strains from the Nordic countries (Norway, Sweden and Denmark) mostly belong to subgroup II [13,19,23]. However, Belgium does not import cattle from those countries. Interestingly, viral strains belonging to subgroup II have been identified in calves vaccinated with vaccine strains belonging to the same subgroup II. These subgroup II viral and vaccine strains are different but very close, with 94 to 96% identity. In that case, an escape from vaccine immunity due to a genetic evolution of the virus is not an option. These results suggest that vaccination in the field faces some practical issues, which can decrease the efficiency of commercial vaccines.

As previously mentioned, several BRSV subgroups have been found in Europe, including subgroups I, II, IV, V, VI, VII and VIII [5,14,15]. While most of our experimental samples were classified into subgroup II or VIII, one sample (i.e., B19.207920) clearly clustered with subgroup III. This is of particular interest, since the subgroup III strains were originally observed in the Americas [5] or, more recently, in Turkey [24] and China [25]. There were no imports of cattle from these countries or their neighboring countries since at least 1989 (unpublished data). According to the breeder, the cattle were vaccinated by intramuscular injection with the MLV-C vaccine, also belonging to subgroup III. Since only one synonymous substitution was observed between this isolate and the MLV-C vaccine strain, a vaccine origin seems most likely.

The subgroup V strains were described as being dominant in Belgium in the late 1990s [5]. However, in our study, we did not detect any isolates that belonged to this subgroup. This observation is also in favor of the shifting of the dominant subgroup of circulating BRSV strains in Wallonia.

BRSV vaccination plays a crucial role in Belgium, particularly given the widespread presence of the Belgian Blue breed, which is known for its limited respiratory capacity. Despite vaccination efforts, veterinarians and farmers frequently encounter cases of mortality among vaccinated cattle, raising concerns about vaccine efficacy. Numerous factors can contribute to vaccine failure, including improper vaccine storage, administration to already infected animals, and suboptimal vaccination protocols. However, the potential for immune escape must also be considered. This study represents a critical first step in addressing this issue, as our findings indicate that the majority of currently circulating BRSV strains are emerging strains, genetically distant from the vaccine strains and previously dominant subgroups.

The four cysteines present in the central conserved region of glycoprotein G of BRSV lead to the formation of a central loop maintained by two disulfide bridges (Cys^173^–Cys^186^ and Cys^176^–Cys^182^) at the top of which is a major epitope site. The amino acids located at positions 177, 180, 183 and 184 strongly contribute to antibody binding directed against BRSV [26]. Mutations at these positions change the local antigenic protein surface and influence the antibody reactivity of BRSV [15,16,26].

As shown in Figure 3, all the field sequences conserved the four cysteines. However, the field strains from subgroup II as well as the vaccine strain B24 belonging to this subgroup had undergone a mutation at position 177 (E^177^ → K^177^) and the strains from subgroups I, V, VI and VII showed a substitution at position 180 (L^180^ → P^180^). All of the strains (field and vaccine) of subgroup II had their leucine replaced by a serine at position 183 (L^183^ → S^183^). These differences may be involved in the pattern of immunogenicity evolution and should be investigated further for the development of future vaccines against BRSV. The subgroup VIII strains did not show mutations at the aforementioned antibody binding sites and 60.9% of the cattle from which these strains were derived were unvaccinated. Future experiments should be performed to determine if the marketed subgroup II and III vaccines are protective against strains of subgroup VIII and the potential role played by the different residues in the cross-protection between subgroups. The mutations found within subgroup VIII (H^162^ → Q^162^, N^163^ → T^163^ and S^165^ → L^165^) were also described in the study by Krešić (2018) [15]. However, these residues do not seem to have an antigenic role or influence the three-dimensional structure of the protein [21].

Analysis of samples from the Flemish region would be necessary to carry out a genetic characterization of the BRSV strains from the whole Belgian territory.

## 5. Conclusions

Our study has provided data demonstrating a clear genetic evolution of BRSV in southern Belgium, with subgroup VIII being the most frequently isolated. This subgroup was not detected in the previous survey conducted two decades ago. The genetic evolution of the virus raises questions about the potential ability of subgroup VIII to evade vaccine-induced immunity. Notably, the commercial vaccines available in Belgium belong to subgroups II and III.

## Figures and Tables

**Figure 1 vaccines-13-00298-f001:**
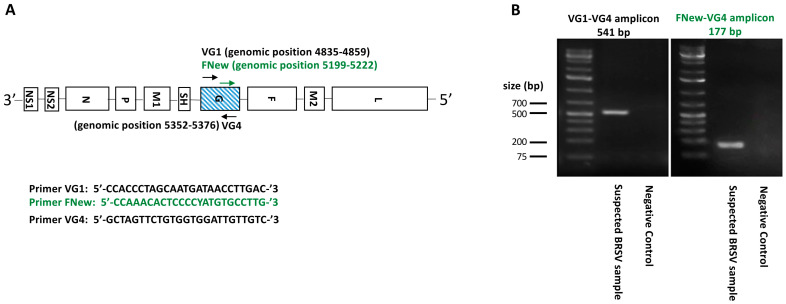
Primer positions and PCR amplicons for the BRSV G gene analysis (**A**). All 51 field isolates and four vaccines were amplified using the primer set VG1 and VG4. To enhance the precision of the 5′ end sequence of the G gene, 11 samples were further analyzed using a second PCR with an additional forward primer: FNew and VG4 (**B**).

**Figure 2 vaccines-13-00298-f002:**
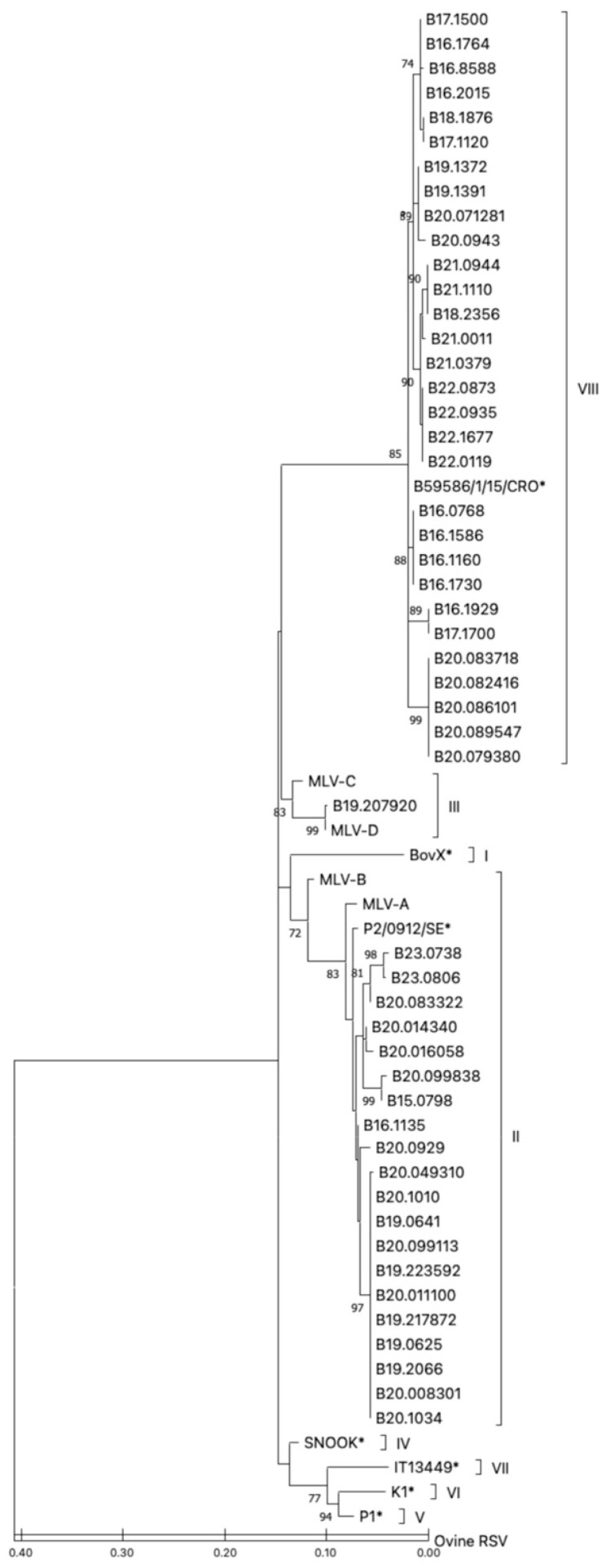
Maximum likelihood phylogenetic tree of the BRSV G gene partial sequences. All 5 field isolates and 4 vaccine strains (MLV-A to -D) were sequenced and trimmed down to a 418 bp segment of the BRSV glycoprotein G ORF and aligned with reference samples from subgroups I–VIII (marked with an asterisk) using ClustalW (Geneious 10.2.6 software, Biomatters Ltd., Auckland, New Zealand). The designations at the ends of the branches refer to the subgroups. Bootstrap values ≥ 70 (1000 repetitions) are indicated by the branch roots.

**Figure 3 vaccines-13-00298-f003:**
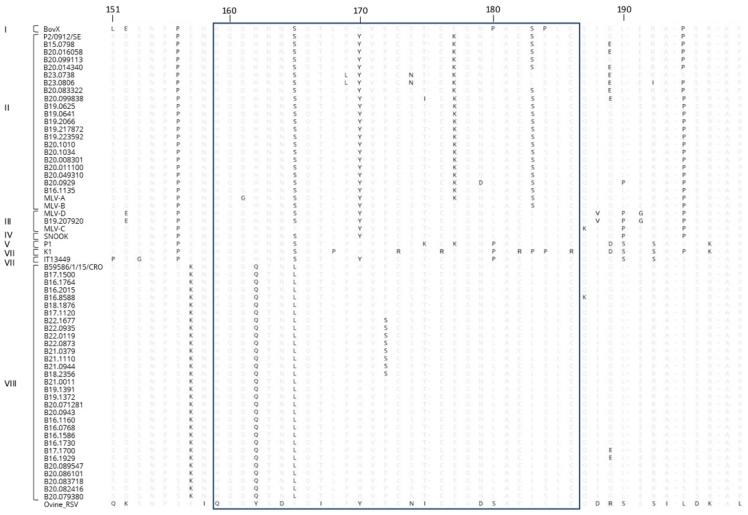
Alignment of the deduced amino acids 151 to 198 of the G gene protein of the BRSV isolates. The central conserved region of the extracellular domain of the BRSV glycoprotein G (positions 159 to 186), considered to be a major immunodominant area, has been boxed. The strains were grouped into the subgroups (I to VIII) shown on the left side.

**Table 1 vaccines-13-00298-t001:** Samples analyzed for the study of bovine respiratory syncytial virus in Wallonia (Belgium).

Subgroup	Isolate Name	Accession Number	Year/Month	Tissue	Age (Months)	Vaccination
II	B15.0798	PP538078	2015/11	Lung	6	No
II	B16.1135	PP538080	2016/01	Lung	2.5	Yes ^b^
II	B19.2066	PP538098	2019/03	Lung	18	No
II	B19.0641	PP538095	2019/12	Lung	6	Yes ^b^
II	B19.0625	PP538094	2019/12	Lung	5	NK
II	B19.217872	PP538100	2019/11	Lung	16	Yes ^b^
II	B19.223592	PP538101	2019/12	Lung	8	No
II	B20.0929	PP538102	2020/02	Lung	4	NK
II	B20.1010	PP538104	2020/02	Lung	7	Yes ^b^
II	B20.1034	PP538105	2020/02	Lung	6	No
II	B20.008301	PP538106	2020/01	Lung	1	Yes ^a^
II	B20.016058	PP538109	2020/01	Lung	4	Yes ^b^
II	B20.011100	PP538107	2020/01	Lung	4	No
II	B20.014340	PP538108	2020/01	Lung	1.5	Yes ^b^
II	B20.049310	PP538110	2020/03	Lung	4	No
II	B20.083322	PP538114	2020/04	Lung	1.5	Yes ^b^
II	B20.099113	PP538118	2020/05	Lung	1.5	Yes ^b^
II	B20.099838	PP538119	2020/05	Lung	3	No
II	B23.0738	PP538128	2023/02	Lung	8	NK
II	B23.0806	PP538129	2023/02	Lung	19	Yes ^b^
II	MLV-A	PP538130	2021/04	Vaccine		
II	MLV-B	PP538132	2021/04	Vaccine		
III	B19.207920	PP538099	2019/11	Lung	7	Yes ^b^
III	MLV-C	PP538131	2021/04	Vaccine		
III	MLV-D	PP538133	2021/04	Vaccine		
VIII	B16.1160	PP538081	2016/01	Lung	5	NK
VIII	B16.1586	PP538083	2016/03	Lung	3	No
VIII	B16.1730	PP538084	2016/04	Lung	2	NK
VIII	B16.1764	PP538085	2016/04	Lung	9	No
VIII	B16.1929	PP538086	2016/05	Lung	2	No
VIII	B16.8588	PP538088	2016/05	Lung	3	Yes ^b^
VIII	B16.2015	PP538087	2016/05	Lung	3	No
VIII	B16.0768	PP538079	2016/11	Lung	4	NK
VIII	B17.1500	PP538090	2017/02	Lung	9.5	Yes ^b^
VIII	B17.1700	PP538091	2017/03	Lung	3	No
VIII	B17.1120	PP538089	2017/12	Lung	5.5	No
VIII	B18.1876	PP538092	2018/04	Lung	4	Yes ^b^
VIII	B18.2356	PP538093	2018/06	Lung	9	No
VIII	B19.1372	PP538096	2019/01	Lung	14	No
VIII	B19.1391	PP538097	2019/01	Lung	6	NK
VIII	B20.0943	PP538103	2020/02	Lung	4	No
VIII	B20.071281	PP538111	2020/04	Lung	1	No
VIII	B20.079380	PP538112	2020/04	Nasal swab	2	Yes ^a^
VIII	B20.082416	PP538113	2020/04	Nasal swab	4.5	No
VIII	B20.083718	PP538115	2020/04	Lung	2	NK
VIII	B20.086101	PP538116	2020/04	Nasal swab	0.5	NK
VIII	B20.089547	PP538117	2020/05	Lung	5	Yes ^a^
VIII	B21.0944	PP538122	2021/02	Lung	4	Yes ^a^
VIII	B21.1110	PP538123	2021/03	Lung	4.5	Yes ^a^
VIII	B21.0011	PP538120	2021/09	Lung	8	Yes ^a^
VIII	B21.0379	PP538121	2021/11	Lung	8	Yes ^b^
VIII	B22.0873	PP538125	2022/02	Lung	2	No
VIII	B22.0935	PP538126	2022/02	Lung	1.5	No
VIII	B22.1677	PP538127	2022/09	Lung	12	Yes ^b^
VIII	B22.0119	PP538124	2022/09	Lung	8	No

Identification country/year of isolation/number, B = Belgium, NK = Not known, ^a^ Calf vaccinated with MLV-A or MLV-B (subgroup II), ^b^ Calf vaccinated with MLV-C or MLV-D (subgroup III).

## Data Availability

Data are contained within the article.

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
