# Peer review of "Phylogenetic Analysis of Bovine Respiratory Syncytial Virus (BRSV) Subgroups in Wallonia Region of Belgium in Relation to Current Vaccination Strategies"

_vaccines, 2025, doi:10.3390/vaccines13030298_

Round 1
Reviewer 1 Report
Comments and Suggestions for Authors
The study presents a phylogenetic analysis of Bovine Respiratory Syncytial Virus (BRSV) subgroups in Wallonia, Belgium, based on genetic sequencing spanning from 2015 to 2023. The study also compares genetic sequences of collected BRSV strains with vaccine strains to assess whether current vaccines are effective. The analysis reveals that circulating BRSV strains belong primarily to subgroups II and III, which align with the available vaccine strains. However, many field strains were identified in subgroup VIII, and no vaccine strain clustered with it, raising concerns about the emergence and spread of subgroup VIII BRSV strains. While subgroup V was previously the dominant strain, this study found no evidence of subgroup V in the analyzed samples. Although current vaccines are based on subgroup II and III, some animals that were vaccinated against these subgroups still became infected with the same strain, suggesting possible immune escape due to mutational changes driven by intensive vaccination strategies.
Table 1 Some results referenced under lines 142 and 156 cannot be found in Table 1. Please add an additional column to include these missing results or provide them in a supplementary table.
Table 1 Ensure that superscripts in the table legend follow an order starting with "a" for the first abbreviation or clarification. The superscript "NK" (Not Known) is not superscript and should be replaced with a different superscript letter to avoid confusion.
Figure 2 Vaccine strains are not clearly defined in the legend. Ensure that vaccine strains are clearly marked in Figure 2. The vaccine strain labeled "MVL" does not match the name used in Figure 2. Please ensure consistency.
Figure 3 The current image quality is poor. Please replace Figure 3 with a high-quality version.
For the methods, please state where samples have been stored during these years and how they have been accessed (thawed or RNA extracted and frozen etc.)
Please explain why Institutional Review Board Statement is Not applicable.
Author Response
Thank you for reviewing this article and for your constructive comments, which have contributed to improving its quality.
Comments 1: Table 1 Some results referenced under lines 142 and 156 cannot be found in Table 1. Please add an additional column to include these missing results or provide them in a supplementary table.
Response 1: A supplementary "Table S1" has been added, providing this information along with additional details on cattle (p12-13).
Comments 2: Table 1 Ensure that superscripts in the table legend follow an order starting with "a" for the first abbreviation or clarification. The superscript "NK" (Not Known) is not superscript and should be replaced with a different superscript letter to avoid confusion.
Response 2: The table legend has been revised, the notation "NK" has been clarified, and it is no longer in superscript.
Comments 3: Figure 2 Vaccine strains are not clearly defined in the legend. Ensure that vaccine strains are clearly marked in Figure 2. The vaccine strain labeled "MVL" does not match the name used in Figure 2. Please ensure consistency.
Response 3: The names of the vaccine strains in Figure 2 have been adjusted for consistency with the text.
Comments 4: Figure 3 The current image quality is poor. Please replace Figure 3 with a high-quality version.
Response 4: Figure 3 has been replaced with a higher quality version
Comments 5: For the methods, please state where samples have been stored during these years and how they have been accessed (thawed or RNA extracted and frozen etc.)
Response 5: The requested information has been added (lines 75–78)
Comments 6: Please explain why Institutional Review Board Statement is Not applicable.
Response 6: The explanations have been added in the appropriate place (lines 332-335)

Reviewer 2 Report
Comments and Suggestions for Authors
This is an interesting and valuable research. However, there are some critical points. There is no “statistical analysis” section in the material and methods, which made it difficult to understand how the data were analyzed. The manuscript is very well-written and well-explained but the authors have to be careful in preparing the manuscript professionally based on the format. please check all the abbreviations carefully. When you introduce an abbreviation for the first time, spell out the full term and follow it with the abbreviation in parentheses. Once you've defined an abbreviation, use it consistently throughout the manuscript. Don't switch between different abbreviations for the same term. Abbreviations stand alone in the “abstract” and in the main text, tables, and figures. and please remove the abbreviations if it only repeat one time. For example, Line 30.BRD and line 34 LRTD were abbreviated but never used in the abstract. Line 45 please define ORF. Moreover, the conclusion is too general with some irrelative statements. please revise based on the achieved results.
Line 150 “two these proportions”
Line 172 “Almost all field isolates were phylogenetically clustered“
Line 279 please remove “This study is the first report of the circulation of BRSV subgroups VIII in Belgium.”
line 280 what does “enhance the need” mean?
Author Response
Thank you for reviewing this article and for your constructive comments, which have contributed to improving its quality.
Comments 1: There is no “statistical analysis” section in the material and methods, which made it difficult to understand how the data were analyzed.
Response 1: A "Statistical Analysis" paragraph has been added in the Materials and Methods section
Comments 2: Check the abbreviations
Response 2: Abbreviations were modified in revised document
Comments 3: Line 45 please define ORF
Response 3: "Open reading frame" was added
Comments 4: the conclusion is too general with some irrelative statements. please revise based on the achieved results.
Responses 4: Conclusion was adapted with more factual sentences
Comments 5: Line 150 “two these proportions”
Responses 5: Modified
Comments 6: Line 172 “Almost all field isolates were phylogenetically clustered“
Response 6: Modified
Comments 7: Line 279 please remove “This study is the first report of the circulation of BRSV subgroups VIII in Belgium.”
Response 7: This sentence was removed
Comments 8: line 280 what does “enhance the need” mean?
Response 8: "Enhance the need" has been replaced with "Underscore the importance of"
Reviewer 3 Report
Comments and Suggestions for Authors
The Editor Vaccines
Thank you for the opportunity to review the manuscript: “Phylogenetic analysis of Bovine Respiratory Syncytial Virus 2 (BRSV) subgroups in the Wallonia region of Belgium”. The paper has been carefully reviewed but significant concerns arose:
This work brings important information about this disease, which can cause significant economic losses. However, the association of different genomes with different clinical presentations, or even the lack of cross-protection between the vaccine and the different genotypes, could be further clarified.
Is there any report of recombination of the vaccine with a field sample?
Author Response
Thank you for reviewing this article and for your constructive comments, which have contributed to improving its quality.
Comments 1: This work brings important information about this disease, which can cause significant economic losses. However, the association of different genomes with different clinical presentations, or even the lack of cross-protection between the vaccine and the different genotypes, could be further clarified.
Response 1: One paragraph was added to the discussion (lines 285-294)
Comments 2: Is there any report of recombination of the vaccine with a field sample?
Response 2: Regarding BRSV and viruses from the Pneumoviridae family, we have not found any reports in the literature describing recombination between vaccine strains and field strains. This may be attributed to the genetic characteristics (non segmented RNA) of these viruses.
Reviewer 4 Report
Comments and Suggestions for Authors
General comments
This study provides valuable and novel data on Phylogenetic characterization of Bovine respiratory syncytial virus among cattle in Belgium. The provided the first report of the circulation of BRSV subgroups VIII in Belgium. The manuscript is well-written and serious English issues were detected.
Abstract
- The authors should indicate briefly the novelty aspects of this study.
Materials and methods
- Reasons of using BRSV Glycoprotein (G) for genetic analysis should be specified.
- Information on sex and animal breed and other information should be added.
Discussion
- Applications of knowledge obtained in this study should be elaborated.
Author Response
Thank you for reviewing this article and for your constructive comments, which have contributed to improving its quality.
Comments 1: The authors should indicate briefly the novelty aspects of this study.
Response 1: We have briefly outlined the innovative aspects of our study in lines 19 and 22-24
Comments 2: Reasons of using BRSV Glycoprotein (G) for genetic analysis should be specified.
Response 2: The reasons have been clarified in lines 97-99.
Comments 3: Information on sex and animal breed and other information should be added.
Response 3: A supplementary "Table S1" has been included (p12-13).
Comments 4: Applications of knowledge obtained in this study should be elaborated
Response 4: A paragraph has been added in the discussion (lines 285-294)
Reviewer 5 Report
Comments and Suggestions for Authors
The study by Demil et al., describes the data on Bovine Respiratory Syncytial Virus (BRSV) subgroups in a limited area (Wallonia region) in Belgium for the several years from 2015 to 2023. The authors conducted sequencing of the isolates and their phylogenetic analysis. Although the study has been done carefully and the manuscript scientifically sounds, I still believe that there are some comments and suggestions, which would significantly improve the manuscript.
- The text does not contain the number and date of the protocol of the ethics committee on these experiments.
- The manufacturers and sources of the four modified-live viral (MLV) vaccine strains marketed in Belgium are not provided. Please insert.
- The study is too focused on a limited region of Belgium, and the manuscript lacks discussion in terms of the phylogeography of Bovine Respiratory Syncytial Virus. In order for this manuscript to be of interest to a wide circle of the scientific community, I recommend expanding the discussion section and adding information on how many isolates are represented in other European countries, which isotypes are predominant, as well as on the representation of the number of isolates in Asia and America.
- Minor: There are some typos scatterplotted throughout the manuscript, i.e. line 131, 277
Author Response
Thank you for reviewing this article and for your constructive comments, which have contributed to improving its quality.
Comments 1: The text does not contain the number and date of the protocol of the ethics committee on these experiments.
Response 1: There is no ethics committee protocol number or date because no animal experimentation was conducted in this study. As clarified in the Institutional Review Board Statement following another reviewer comment "All tested lung samples from deceased animals were delivered to our laboratory following necropsy to answer the diagnostic need of field veterinarians. In the same way, nasal swabs were collected as part of routine respiratory disease diagnostics in farms. No other animal or dedicated samples were involved in this study." (Lines 332-335)
Comments 2: The manufacturers and sources of the four modified-live viral (MLV) vaccine strains marketed in Belgium are not provided. Please insert.
Response 2: The names of the vaccines have been added in the introduction; however, we have not indicated the specific correspondence in the text, as we did not receive authorization from the pharmaceutical companies. Moreover, since there is no significant difference between the vaccine strains, this information would not contribute to the discussion.
Comments 3: The study is too focused on a limited region of Belgium, and the manuscript lacks discussion in terms of the phylogeography of Bovine Respiratory Syncytial Virus. In order for this manuscript to be of interest to a wide circle of the scientific community, I recommend expanding the discussion section and adding information on how many isolates are represented in other European countries, which isotypes are predominant, as well as on the representation of the number of isolates in Asia and America.
Response 3: Informations was added to the discussion (Lines 226-236)
Comments 4: There are some typos scatterplotted throughout the manuscript, i.e. line 131, 277
Response 4: The reference indicated in superscript has been corrected (line 131). A full stop has been added at the end of the sentence (line 277)
Round 2
Reviewer 1 Report
Comments and Suggestions for Authors
Authors have improved the quality of images and provided additional information I have requested. Thanks for addressing my comments.
Comments on the Quality of English LanguageEnglish language quality is fine and understandable.
Author Response
Authors have improved the quality of images and provided additional information I have requested. Thanks for addressing my comments.
Answer: Thanks for reviewing and formulating suggestions to improve our document.
Reviewer 2 Report
Comments and Suggestions for Authors
-
Author Response
Thank you for your comments, which contributed to the improvement of the document